# Importance of Utilizing Non-Communicable Disease Screening Tools; Ward-Based Community Health Care Workers of South Africa Explain

**DOI:** 10.3390/ijerph21030263

**Published:** 2024-02-24

**Authors:** Elelwani Malau, Irene Thifhelimbilu Ramavhoya, Melitah Molatelo Rasweswe

**Affiliations:** 1Department of Nursing Science, University of Pretoria, P.O. Box X323, Pretoria 0007, South Africa; malau.elelwani@gmail.com; 2Department of Nursing Science, University of Limpopo, P.O. Box 1106, Polokwane 0727, South Africa; melitah.rasweswe@ul.ac.za

**Keywords:** ward-based community health care workers, screening tools, utilizing, non-communicable diseases

## Abstract

The screening of patients in the community is important and is a commonly used indicator to detect, prevent, and treat abnormal health changes. As such, the South African Department of Health following the initiative of the World Health Organization has appointed ward-based community health care workers through a primary health care reengineering program. The main objective of their appointment was to screen household members to reduce the burden of diseases such as diabetes mellitus and hypertension. As such, the study investigated the importance of using non-communicable disease screening tools by ward-based community health care workers in South Africa. A qualitative, exploratory, and descriptive design was used. A non-probability purposive sampling method was used to select forty participants from primary health care facilities. Four focus group discussions were held with ten participants in each group. Semi-structured focus group discussions were held with participants in their workplaces. Content data analysis was applied to come up with one theme and six subthemes. The study findings revealed that the use of screening tools facilitated comprehensive household assessments, helped identify risk factors and symptoms, and facilitated health education and patient referrals. The continuous supply of screening tools and updates on their use was recommended to reduce the rate and burden caused by non-communicable diseases to society at large.

## 1. Introduction and Background

Non-communicable diseases (NCDs) account for a greater proportion of deaths worldwide each year [1,2]. About one-fourth of global NCD-related deaths take place before the age of 60 with 80% of NCD deaths occurring in low- and middle-income countries [3,4]. Of the 57 million deaths that occurred globally in 2008, 36 million which is almost two-thirds were due to NCDs [2]. According to the World Health Organization, 17.9 million people died from cardiovascular illnesses in 2023, and 2.0 million were due to complications of hypertension, while 2.0 million were from diabetes.

The combined burden of these diseases is rising fastest among lower-income countries, disadvantaged populations and communities, where they impose large, avoidable costs in human, social, and economic terms [5]. 

As a result, the World Health Organization (WHO) planned the package of essential non-communicable disease interventions for primary health facilities. To implement the package, an engagement was organized with local community facilitators for the training of community health workers (CHWs) as the package was designed for their use [3]. The training was designed to link the health service and the community to meet the target of at least 50% of eligible individuals receiving treatment and counseling through early detection. This aids in the management of many illnesses, particularly non-communicable diseases. Numerous studies from China, Brazil, and Iran attest to the management of a variety of diseases, moreover, the WHO [6] recommended the employment of community health workers to assist in improving the health outcomes of the population they serve.

Based on the above findings of the WHO and the experiences of other countries, South Africa (SA) adopted the primary health care reengineering framework in 2011, and a strategy using ward-based primary health care outreach teams (WBPHCOTs) [7]. Through these teams, SA seeks to ensure universal health coverage of its people and to improve the quality of primary health care services [8]. The teams are based in primary health care facilities and offers integrated services to households and individuals within its catchment area. The catchment area refers to wards (group of villages) which were demarcated politically, being governed by an elected counselor. WBPHCOT was developed in preexisting structures that were then called community home-based care, established by a non-government organization to substantiate the care provided by primary health care nurses [9]. The structure provided minimal health care in the homes of the community, which included the promotion of environmental and personal hygiene, the care of patients suffering from chronic conditions, and their nutritional status [8]. Since the introduction of WBPHCOTs, some of the community care members were appointed as ward-based community health care workers (WBCHCWs) and trained to serve and support the distribution of primary health care services within the context of national health insurance. Ward-based community health care workers provide primary healthcare services to families/households; community outreach services; preventative, promotive, rehabilitative, and palliative services in the wards (villages). They were trained for 59 days. Their training included screening of communicable and non-communicable diseases (diabetic mellitus, hypertension, tuberculosis, and human deficiency virus), distribution of medication, checking adherence of patients to treatment, conducting follow-ups, and tracing of defaulters. It also included the identification of pregnancy and postnatal visits [7]. Despite the training of WBCHCWs, an analysis of the data from other studies revealed that WBPHCOTs do not function optimally, and the implementation of the strategy has been highly uneven throughout the country [8,10]. In order to oversee and supervise the work of WBCHCWs, a team leader who is a licensed professional nurse was appointed in January 2012. Since then, the WBCHCWs have been reporting their activities, tasks, and actions via the District Health Information System (DHIS) [7,11]. The main objective of the WBCHCWs appointment was to perform screening in the households of their communities and to identify and detect patients at risk of preventable non-communicable conditions. The idea was to reduce the burden of diseases that include diabetes mellitus and hypertension, which was the focus of the current study. According to the study conducted by Tatsumi and Ohkubo [12], approximately 50% of patients suffering from hypertension develop diabetic mellitus and as such, it accounts to 40% of patients suffering from it globally [3,13]. In 2016, the Demographic and Health Survey indicated the prevalence of hypertension in South Africa to be at 48.2% [14]. The rate of diabetic mellitus has risen from 4.5% in 2010 to 12.7% in 2019 [15]. Furthermore, Statistics South Africa [16] reported that over a 20-year period, the number of deaths from non-communicable diseases such as diabetes and hypertension rose by 58.7%, from 103,428 in 1997 to 164,205 in 2018. For men and women, respectively, the median age of death in years was 65 and 69 [16]. In the province of Limpopo, where this study was carried out, over 150 individuals receive their monthly treatment for hypertension and diabetes. Two to three adults and children with the two signs of illness and symptoms are seen each day, some referred by WBCHCWs, and monthly more than three are newly diagnosed with the two conditions [17]. As such, the current study was conducted with the objective of investigating the importance of using non-communicable disease screening tools by WBCHCWs of the Fetakgomo-Tubatse subdistrict, Limpopo Province, South Africa.

## 2. Materials and Methods

### 2.1. Study Design

In the current study, a qualitative exploratory descriptive research design was used to gather in-depth information from WBCHCWs on the importance of utilizing non-communicable disease screening tools for diabetic mellitus and hypertension.

### 2.2. Study Area

The study was carried out in Limpopo Province, which is one of the rural provinces located in the north-east corner of South Africa. The Limpopo Province has a population of 6.015 million people and 1.64 million households [18]. Limpopo Province has five districts consisting of Capricorn, Vhembe, Mopani, Waterberg, and Sekhukhune, with 22 subdistricts. The Sekhukhune district consists of Elias Motsoaledi, Ephraim Mogale, Makhuduthamaga, and the Fetakgomo-Tubatse subdistricts. In Fetakgomo-Tubatse, there is Burgersfort Clinic, Mecklenburg Gateway Clinic, Selala Clinic, HC Boshoff Health Centre, Dilokong Gateway Clinic, Madiseng Satellite Clinic, and Riba Clinic. This subdistrict was selected because it has consistently retained the WBCHCWs since 2012 and has larger household population (125,361) compared to other subdistricts. In the current study, only three clinics from the Sekhukhune district in Fetakgomo-Tubaste were selected: Dilokong Gateway Clinic, Madiseng Mobile Satellite Clinic, and HC Boschoff Clinic.

## 3. Population and Sampling

Population is defined as an entire group of persons or objects that are of interest to the researcher, in other words, those who meet the criteria of the study being conducted [19,20,21]. The populations in the current study were all trained WBCHCWs utilizing non-communicable disease screening tools for hypertension and diabetic mellitus when assessing households’ members from Fetakgomo-Tubatse subdistrict, Limpopo Province, South Africa. The non-probability purposive sampling method was chosen based on the researcher’s knowledge of the population and was used to select participants from the three clinics at Fetakgomo-Tubatse subdistrict, Limpopo Province. The sampling method implies that there is no way to ensure that each member of the population could be selected although participants were selected based on the judgment of the researcher and their involvement in the screening of households’ members using the non-communicable tool [21,22]. Therefore, the current study selected forty (40) trained WBCHCWs purposively to be part of the focus group discussion.

## 4. Data Collection and Analysis

Burns and Grove [20] define data collection as a precise and systematic gathering of information relevant to a study, as such, with the current study, data collection was carried out immediately after the researchers obtained the consent of the participants. It was collected for 3 weeks between March and April 2022 in the designated clinics where participants are registered, meaning their natural setting. The researchers made appointments with the participants and agreed on the time and date for each group to meet. Researchers used the focus group discussion (FGD) method to collect data from participants. The focus group is a type of group interview, exploring a range of certain topics, since they entail some sort of group activity [23,24]. Focus group interviews were found to be important in this study because our goal was to gather a significant number of qualitative data based on the research question. It also assisted in exploring collective perspectives, attitudes, behaviors, and experiences of WBCHCs. The interviews during a focus group allow interactions between participants, hence it yields rich and meaningful data [25]. Four focus group interviews were conducted in three clinics: two in clinic C and one in clinic A and B, respectively. As suggested by Brink et al. [19] and Grove et al. [20] 5–12 participants are ideal for a focus group, and so in our study, the number of participants per focus group was 10, in order for the groups to be manageable. The time range of the focus group interviews was between 1 h 30 min and 2 h, depending upon the number of individuals in attendance, and the intensity of the discussion. Participants in each FGD were chosen based on their location of residence and kind of employment. Additionally, to enrich understanding and provide alternative insights [26], field notes on participant interactions were documented. In the current study, forty (40) WBCHCs who were utilizing non-communicable disease screening tools to screen patients in their respective homes were selected to be part of the focus group interviews. The principal investigator (researcher) facilitated the FGD, and ground rules were set before the discussion. Participants agreed to raise a hand if they want to explain the concept under investigation, to respect each other, and give each other a chance to state their case and to participate actively. A code was assigned to each participant according to the focus group interview number, in order to avoid mentioning their names, for example FGD1 P1. The FGDs were held in a private place arranged with participants in their respective allocated clinics. The place was conducive for the interviews and free from distractions. The researcher asked one central question to the group which was the following: ‘*What is the importance of using non-communicable screening tools based on diabetic and hypertensive conditions within households?*’ The participants were allowed time to process the question and were spontaneous in responding. Similar to the study conducted by Kitzinger [23], complementary interaction was emphasized in the current study as such, participants deliberated on the question asked from one participant to another supporting each other’s answers when indicating what they knew as the importance of using the non-communicable disease screening tools. From their engagement, probing questions were asked to obtain more detail and clarity based on their responses. A tape recorder was used to record the responses of the participants with their consent and the WBPHCOT team leader assisted the researcher in writing field notes. Data were collected on various days until saturation was reached at FGD number three; however, the fourth FGD was conducted to confirm that there is no new information coming. For data analysis, eight steps of content analysis were applied [27]. The principal researcher transcribed the data verbatim and read the participants’ narratives to acquire a feeling of their ideas and fully understand them. The transcription of data included the non-verbal clues as observed from participants. This was followed by categorizing similar data and assigning codes. This process was repeated until all the collected data were coded. The principal investigator submitted data to other researchers to recheck the narratives and coded data to ensure consistency. Lastly, researchers submitted the coded data to the independent co-coder to recheck the analyzed data together with the transcript to confirm the codes. In the final step, consensus was reached between the independent co-coder and researchers regarding the coded data, resulting in one theme and six subthemes.

## 5. Measures to Ensure Trustworthiness

Trustworthiness was ensured through credibility, transformability, and dependability. To ensure credibility, prolonged participation, persistent observation, and peer debriefing were applied. The researcher invested enough time, established trust in the WBCHCWs, kept the investigation honest, and identified elements relevant to the study. In the current study, field notes and transcriptions were given to an independent coder who had not participated in the study to confirm the analyses. The transcribed data were handed to the supervisor to re-check and ensure that all findings were supported by data from the participants. Audit recordings were helpful as part of the audit trail and served as a reminder of what transpired between researchers and participants. The researcher provided enough information on how the study unfolded, the methods used to sample the participants and the design, as well as how the data were analyzed for other readers to ensure that they were transferable to other studies if a similar study can be conducted in a context like the current study.

## 6. Ethical Considerations

Ethical consideration was obtained from the Research Ethics Committee of the Faculty of Health Care Sciences, University of Pretoria no. 167/2020 and the Research Committee of the Department of Health of Limpopo Province. The researchers also received permission from the managers of the selected clinics. Before starting the focus group discussion, the researcher gave verbal information to the participants, followed by written informed consent. To avoid pressuring the participants, they were told to withdraw from the study at any time without prejudice. To ensure anonymity of the names of three clinics and of participants, codes were assigned, and as such, clinics were indicated as Clinic 1, 2, and 3 and participants as 1, 2, and so forth. All participants responded anonymously according to the code assigned to them.

## 7. Results

Table 1 below indicates one (1) theme with six (6) subthemes that emerged from the current study.

## 8. An Outline of the Importance of Utilizing Non-Communicable Disease Screening Tools

Participants reported that screening tools facilitate a comprehensive household assessment. They have identified specific disorders, risk factors, and symptoms with screening tools that required health education and referral to primary health care services and other members of the multidisciplinary team for continuity of care. Community and household members were made aware of the potential health problems identified. The use of screening tools promoted trust among the community and participants and evidence of their contributions to society was observed. The use of screening tools guides the actions and workflow of WBCHCWs and improves the client’s compliance with the advice offered. The following subthemes emerged as indicated in Table 1 above:

### 8.1. Facilitates a Comprehensive Assessment of the Household

The WBCHCWs are assigned 270 households to screen and monitor individually yearly. During the screening process for family members, they have reported that in addition to screening for diabetes mellitus and hypertension, they were able to identify families who experienced lack of nutritious food in the household that predisposes family members to other conditions.

FGD1-P2: “when we have entered a household asking them questions regarding diabetics and hypertension, one gets to know that in this home they do not have enough nutritious foods”.

Other participants added and reported that, in addition to those who did not have enough nutritious food, they recognize sick family members who are unemployed. They referred them to dietitians and social workers who help them with food packages.

FGD3-P7: “…when we screen for households, apart from identifying lack of food in their homes, sometimes we find members of the community who are unemployed and sick, so we refer them to social workers and dieticians for help”.

Participants added and indicated that they develop a good relationship with community members; as such, they feel free to talk about all the problems of every patient in their homes. They report information on the number of sick and vulnerable patients at home.

FGD2-P4 and P10: “Members of the household become open when one asks them questions using the screening tool because they can see that we are not only focusing on them alone. The family members tell us about everything. Sometimes as a community health care worker, you would like to know how many people are living in that family and the number of those who are sick and those who are in need? Screening tool helps us with this kind of information”.

The findings of the current study showed that the use of a screening tool has a positive impact on both community health workers and community members. Community members can be open and express their feelings about their disease and share their problems with WBCHCWs at home. The WBCHCWs can identify those who are sick and needy there by referring them to other members of the multidisciplinary team for further assistance.

### 8.2. Facilitates the Identification of Risk Factors and Symptoms for Referral

Most of the participants indicated the value of using the screening tool, as it facilitated the identification of risk factors and symptoms that require referral. Screening patients helped them understand the extent of the problems of their patients.

FGD1-P6: “another thing you can understand is if the person might have diabetes or not? Then you can refer the person to the clinic so that they can be sure if it is diabetes or not. I may assume that the patient might have diabetes based on his signs, I then refer him to the clinic or hospital to be sure of his condition”.

FGD3-P8: “Yes, I see that it is very significant to use the screening tool, we can distinguish the diseases of our patients, so I can know if this needs to be referred or not”.

One of the participants added and indicated that these prevent guessing what the problem is and provide the correct information for proper decision making.

FGD2-P11: “I see that the screening tool is useful because you will find that someone in the household may have signs but not knowing what the problem might be. So, when you have the screening tool reading for that person, he/she will be helped and the person ends up telling you that you see signs 1, 2, 3. Then you can talk to that person and tell him to go to the clinic to be checked. Because we know since we are working with people that when he/she tells us that maybe it is high blood because he feels dizzy, etc. The only thing you can tell him is to go to the clinic so that he can know what his real problem might be”.

The findings of the current study showed that screening helped participants understand what the person was suffering from, so they can decide whether to refer to the clinic or not.

### 8.3. Facilitation of Health Education

Most of the participants indicated that the use of screening tools facilitated the health education of household members about signs of diabetes and hypertension; it guided them on how to read the results after monitoring their diabetes and hypertension status. In addition, they created awareness of the two conditions and advised members on healthy eating styles such as exercise, diet, and a suitable diet to reduce the risk of the two conditions.

FGD2-P12: “They are immensely helpful to us, especially when we are in the field asking patients some questions and ticking. It also helps to teach about diseases related to diabetes and hypertension”.

FGD2-P14 and FGD3-P2: “That is true, screening tools help us a lot as we can identify signs of diabetes or high blood pressure. If we find them, we can teach household members what type of food they must eat and even encourage them to do exercise”.

One of the participants added and indicated that the screening tools help to raise awareness.

FGD2-P4: “You find that my diabetic patient still has a problem of eating too much, so you must inform him through education that a diabetic patient does not eat too much food at the same time”.

FGD3-P6: “Patients must eat a small amount of food, but often do not eat an excessively large porridge because that will cause a problem in their body.”

It was evident that the use of screening tools assisted participants in educating household members about non-communicable conditions and how to prevent them from occurring.

### 8.4. Facilitates the Acceptance/Awareness of Clients of Potential Health Problems

Many participants indicated that patient screening tools facilitated the awareness and acceptance of potential health problems by their clients. Participants indicated that the screening tool facilitates awareness of health risks and their chronic condition because once patients know their health conditions, they change their lifestyle.

FGD1-P3: “I think this screening tool, according to how the questions are written, helps me to identify problems from the clients, for example, if a person is a smoker and has hypertension, following the sequence of questioning a client, I am able to conclude, that there are some health risks and refer the client to the health care facility”.

Another participant explored the issue further by indicating the following:

FGD1-P12: “I can detect risk factors with the use of the screening tool, if I visit a household, and start asking questions guided by the screening tool, I tick accordingly and at the end I see if the family or any of the family members is exposed to risk factors of illnesses’’.

In addition to what participant 12 reported, other participants indicated that screening helps the community to know whether they have hypertension or diabetes.

FGD1-P1: “What I was able to experience or see is that this screening tool helps us to be aware of the people who are suffering from hypertension or diabetes in the community. Sometimes, a person may have hypertension or diabetes without realizing it. So, according to the signs when I explain them to him, the signs for diabetes are this and that for high blood are this and that. Then a person ends up realizing that he/she has a problem, then went to the clinic to seek medical services on time or before complications”.

FGD3-P4: “The screening tool helps the community members to be aware of the signs of hypertension and diabetes, and not take them lightly, they accept that there might be a problem and seek health care assistance in the hospital or clinic”.

The current study found that the screening tools are appreciated by the WBCHCWs because they can conclude if the clients are at risk of developing or suffering from hypertension or diabetes mellitus.

### 8.5. Guides WBCHCWs’ Actions and Workflow

Most of the participants reported that the use of patient screening tools is important, because it guides their actions during households visit. The tools made their fieldwork simplified and easier. Participants indicated that screening tools helped them differentiate signs and symptoms of various conditions, including hypertension and diabetes mellitus.

FGD1-P4 and FGD4-P2: “This screening tool also helps us when we are talking to the community members, we eventually know what signs might be for. As we will be checking the tool while the patient speaks, we will know how to assist him”.

Other participants agreed and indicated the following:

FGD1-P11: “True, also nodding her head, when I use this screening tool, it makes my work easier, when I visit a household, I know what to do, because it directs me, the way questions are constructed allow the family members to be open and provide answers”.

FGD2-P11: “I see it being useful, as it simplifies our work. Like they already said, when you go from house to house having it, you can save time. Because you just read and the patient gives you an answer, you just tick and analyze at the end, and know what to do if the patient answered this way”.

The participant indicated that the screening tools reduce workloads as they work efficiently.

FGD4-P7: “The hypertension and diabetes screening tool are very important to us as community health workers, it reduces our workload because we asked all the questions about signs and symptoms while referring to the screening tool, and I immediately know if the person is healthy or suffering from any of diseases as directed by the screening tool”.

Therefore, the screening tools were found to be important in guiding the work activities that the WBCHCWs are expected to carry when visiting households.

### 8.6. Enhances the Compliance of the Clients

Most of the participants indicated positive experiences with the use of patient screening tools, as they helped guide patients on how to take their medications to ensure compliance and adherence. The importance of taking medications regularly and on time was stressed. Screening tools also helped to teach patients how to manage the side effects of their medication.

FGD2-P6: “It also helps us to educate clients on how to take their medication for it to work effectively”.

FGD2-P12: “Yes …… the screening tools emphasize how important it is to take medications as prescribed and to avoid non-adherence. This is because hypertension and diabetes are chronic diseases that last for life, which is what we teach them when we go to the field”.

FGD2-P5: “I add to the information about the medications, we advise them to take their treatment on time, for example, if the patient is used to taking treatment at 7:00, he should take treatment at 7:00 if it is 08:00 or 09:00, he should take it at the same time, some medications are taken in the morning and some at night”.

Participants indicated that screening tools helped teach patients how to manage the after-effects of medication.

FGD3-P2: “If the patient says that the medication is making them dizzy. As a WBCHCW, I would start asking questions. If the patient ate food before taking medication or not, and check if the medication is taken before or after the medication. Then I advise on the correct measures to promote adherence. If it is a side effect, the screening tool suggests referral to the health care facility”.

The study revealed that screening tools were used as a guide for the WBCHCW to teach patients on how to take medication, to promote adherence and prevent defaulting. Additionally, screening tools provide the participants with content of educating patients about drug side effects and what to do in case they develop.

## 9. Discussion of Results

As indicated in Table 1, the findings of the current study revealed that the use of patient screening tools facilitates comprehensive household assessments, the identification of risk factors and symptoms that require referrals, the identification of specific disorders, health education, client acceptance/awareness of potential health problems, and promotes trustworthiness and evidence of the contributions of WBCHCW.

### 9.1. Facilitates a Comprehensive Household Assessment

The findings of the current study showed that the use of the screening tool had a positive impact on WBCHCWs, households, and community members. The members of the community and the households were open and expressed their feelings about their illness and shared their health problems with the WBCHCWs. Ward-based community health workers were able to identify the problem and the disease during household visits that required them to intervene and act by referring the patient to the multidisciplinary team depending on the problem and condition of the patients, for example, to social workers, dietitians, clinics, and hospitals. To support the findings of the current study, Perry and Zullige [28] reported that WBCHWs are the world’s most promising health workforce as they use screening tools to identify various non-communicable diseases with the aim of reducing the burden of disease from serious, easily preventable, or treatable conditions. The authors also stated that they are the first point of contact between health service clients and providers, linking communities to the health system [28].

The findings of the current study revealed that the WBCHWs were able to identify other households’ problems, such as lack of nutritious food, which could lead to other conditions. As such, Khuzwayo and Moshabela’s [29] study highlighted that when WBCHWs use the screening tool, it helps to detect other problems within the family, including lack of food items and other members who are affected by other diseases. As such, the use of screening tools that facilitate comprehensive assessments will assist in early detection of other conditions that could facilitate early referrals to the next level and for the required interventions of other multidisciplinary teams.

### 9.2. Facilitates Identification of Risk Factors and Symptoms for Referral

The findings of the current study revealed the use of patient screening tools that facilitated the identification of risk factors and symptoms for referral. Screening patients with screening tools helps WBCHWs understand the extent of family member problems so that they can refer clients to the clinic or hospital for further investigations. Participants indicated that screening helps them understand what the person is suffering from and can decide whether to refer or not and prevents making assumptions about the patient’s condition. Using screening tools prevented guess work on what the problem is and provided correct information for proper decision making.

The study findings were supported by the World Health Organization [30], which stipulated guidelines for the performance of community health workers, as they were to refer people with presumptive signs and symptoms of various conditions for diagnosis and related treatment to hospitals or clinics. The effectiveness of WBCHCWs to pre-screen people for high-risk factors and other conditions is a first step toward subsequently increasing screening that requires a high level of referral [28]. In line with the study findings of Tshikombana and Ramukumba [31] screening tools enable them to identify signs and symptoms of diseases and refer patients accordingly. Like the study by Scott et al. [32] the researchers indicated that WBCHWs assist with appropriate use of screening tools and make referrals to the next level. Referrals to the next level of care will help patients receive prompt treatment and care, and as such, complications are prevented. Various authors agreed with the findings of the current study where the authors indicated that conducting a household evaluation helped community health workers identify other risk factors related to conditions such as cancer in addition to diabetic and hypertensive diseases [32,33].

### 9.3. Facilitates Health Education

The findings of the current study revealed that the use of the screening tool facilitated the need for health education. It was evident that the WBCHCWs guided family members on how to read the results after checking and recording their blood glucose and blood pressure. The WBCHCWs gave family members advice on a healthy lifestyle such as exercise and diet, and educated the community about the signs of diabetes and hypertension so that they could be aware and act early. According to the results of the current study, the World Health Organization [34] suggested the provision of health education by WBCHCWs as they are the first point of contact. Similarly, to the findings of the current study, Schneider et al. [9] indicated that WBCHCWs facilitates health education for families by teaching about diabetic mellitus and hypertension as a condition. The authors also highlighted the use of screening and health promotion programs in schools and early childhood development centers, working in partnership with school health teams and outreach teams to educate school children and their families about chronic conditions [9]. Moloko and Ramukumba [35] supported that WBCHWs participate in various health promotion campaigns, organized by the health facility, these involved screening for TB, cancer, male circumcision, physical activity for the elderly, adherence clubs, and school health promotion. Giving health education can assist the community and family members, as they will have knowledge of various diseases and how to prevent them from occurring, thus increasing their life expectancy.

### 9.4. Facilitates Clients’ Acceptance/Awareness of Potential Health Problems

This study revealed that the use of patient screening tools facilitated acceptance/awareness of potential health problems by household members. Awareness was based on the identification of risk factors, signs of diabetes, and hypertension by the WBCHWs, which assisted patients in taking measures to change their lifestyle and thus reduce the risk of suffering from the conditions. The findings of this study were supported by the World Health Organization [34], which indicated that the screening of patients by health care professionals through the taking of history facilitated awareness of the signs and symptoms the patient present. When the health care providers are aware of what the patient is suffering from, they can advise the patient, who in turn accepts the diagnosis [36,37]. The same authors further highlighted that once the diagnosis is accepted, there will be behavioral changes regarding adherence to treatment and other healthy behavior because one of the main advantages of advanced medical treatments in today’s health care environment is the ability of the patients and their families to be aware of their condition, understand health, and accept medical information. Therefore, acceptance and awareness of potential problems around families detected through screening tools can help them change bad habits that can cause hypertension and diabetes mellitus, for example, a smoker will stop smoking to prevent the worsening of hypertensive and diabetic mellitus.

### 9.5. Guides WBCHCWs’ Actions and Workflows

The WBPHCOT program plays a critical role in extending PHC services to the community and household level and making health accessible in terms of distance and information. The program assisted WBCHCWs’ actions and workflow through the provision of screening tools. As such, this study revealed the positive experiences of WBCHCWs in using patient screening tools that guided them to understand the signs and symptoms of chronic diseases and to simplify and amplify field work. The screening tool used by WBCHCWs directs health care teams to promote good health and prevent disease through a variety of interventions [38]. A healthy community, healthy family, healthy individual, and a healthy environment will result from this. According to the participants in our study, the screening tools made it easier for them to ask clients pertinent questions and served as a manual for the WBCHCW team to carry out their responsibilities effectively. Similar results were reported in Mpumalanga province, South Africa [38]. Ormel, Kok, Kane et al. [39] have suggested that WBCHCWs should be supported by the health system and provide them with resources, including screening tools, to perform their duties effectively. Vaughan, Kok, Witter, and Dieleman, [40] were further supported by indicating that WBCHCWs deliver a wide range of promotion and preventive services by using a screening tool. The WBCHCWs are also in the unique position of being able to bring greater knowledge about the health of the community to the next level. Therefore, the use of screening tools as a guide for WBCHCWs’ workflows and actions is crucial and recommended.

### 9.6. Enhances the Compliance of the Clients

This study revealed the importance of using non-communicable disease screening tools, as they informed patients on the importance of compliance and adherence to their medication, as well as the management of side effects. According to the results of the current study, screening tools helped to provide patient education on how to take both hypertension and diabetic mellitus treatment, and on how to manage side effects of the drug, just like in the study that explored the perceptions and experiences of taking oral medications for the treatment of type 2 diabetes mellitus [41]. The same authors also recommended that to improve linkage with patient treatment, support should be offered when initiating treatment to maintain patient outcomes, including treatment adherence [41]. Engaging clients in their treatment and offering support through follow-ups as performed by WBCHCW in our study promotes patient compliance to treatment and a greater degree of treatment satisfaction [42,43]. Furthermore, Soleymani and Wallace-Bell [44] provided evidence to bolster the conclusions by showing a correlation between stakeholders heightened participation and improved treatment outcomes. As screening tools enhance client medication compliance, they will further assist in the prevention of complications of hypertension, diabetes mellitus, and other diseases. Thus, promoting the quality of patient lives.

## 10. Study Limitation

The study followed qualitative, exploratory, descriptive research designs, and as such, the findings cannot be generalized to other settings, as only three clinics were used as the study sites. However, the findings of the current study can be of benefit to the Limpopo department of health since the screening tool assisted in the identification of patients with signs and symptoms of hypertension and diabetes mellitus. As such, the use of screening tools by WBCHCWs assisted the community members to seek health care services on time; although, the researchers cannot confirm if all sought medical assistance. There is no scientific evidence indicating patients who adhered to treatment nor change their lifestyle as advised by WBCHCWs. Another limitation to this study was lack of evidence of those who were referred to the health care facilities with signs and symptoms of the two conditions, if all adhered to their referrals, and if their diagnoses were confirmed. The study was conducted in one subdistrict due to lack of funding; the findings would have been different if other subdistricts participated.

## 11. Conclusions

In this study, it was evident that the screening tools helped WBCHCWs as they facilitate comprehensive household evaluations whereby WBCHCWs had identified other problems in addition to hypertension and diabetic conditions. Furthermore, risk factors and symptoms of diabetic and hypertensive conditions were identified from other family members who did not suffer from the two mentioned conditions that required referral to the clinic and other multidisciplinary teams. After screening, the WBCHCWs provided health education to household members according to the risk factors identified. Members of the household were informed of the risky lifestyle behaviors that lead to hypertension and diabetic mellitus and accepted their conditions by adhering to a healthy lifestyle. Furthermore, the screening tools promoted trustworthiness between WBCHCWs, families, and the community, as they provided evidence of the contributions of WBCHCWs. Guided by the screening tools, the actions and workflow of WBCHCWs became simpler and easier. The continuous supply of screening tools and updates on their use was recommended to reduce the rate and burden caused by non- communicable diseases to society at large.

## Figures and Tables

**Table 1 ijerph-21-00263-t001:** Theme and subthemes of WBCHCWs regarding the use of screening tools.

Themes	Categories
An outline of the importance of utilizing non-communicable disease screening tools	Facilitates a comprehensive assessment of the household.
	Facilitates the identification of risk factors and symptoms for referral.
	Facilitates health education.
	Facilitates client acceptance/awareness of potential health problems.
	Guides WBCHCWs’ actions and workflow.
	Improves client compliance.

## Data Availability

Appendix A for the findings of this research are available on request.

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
