# Peer review of "Importance of Utilizing Non-Communicable Disease Screening Tools; Ward-Based Community Health Care Workers of South Africa Explain"

_ijerph, 2024, doi:10.3390/ijerph21030263_

Round 1

Reviewer 1 Report

Comments and Suggestions for Authors

Some clarifications are needed, as pointed out in the attached PDF edited.  

Comments on the Quality of English Language

Some clarifications needed as indicated in attached.  

Author Response

Good day

Thank you for taking your time to review our paper, here are the comments and revised paper.

AUTHOR’S CORRECTIONS TO REVIEWER 1 ‘S COMMENTS

Date: 31/01/2024

Title: Importance of utilizing noncommunicable diseases screening tools; Ward based community health care workers of South Africa Explains.

Reviewers’ comments

Author’s corrections

SHORTER TITLE SUGGESTED

 Community health workers in South Africa ward basedut screening for non-communicable diseases

Suggestion welcomed, but the suggested topic is not in line with the objective of the study

Abstract

-delete caused by quadruple

-Spell it out before FGDs and define Ward

-Quadruple deleted, line 13 a

-Focus group discussion was indicated in full, and Ward stands for villages or communities being grouped together, line 58

Population and sampling

Give some details of training eg duration, content

 Details of training was indicated on introduction line 71-74

Data collection

Stationed replace with registered

Registered written

Measures to ensure trustworthiness.

To ensure credibility, prolonged participation, persistent observation, and peer de-briefing were applied- Applied was indicated as not clear

Applied changed to implemented

Results

Most of the participants indicated the value of using the screening tool

Participants refers to ward-based community health care workers as they were our participants.

One of the quote indicated that “another thing you can understand is if the person might have diabetes or not, the question from the reviewer was; Do you mean, were they told by a doctor that they had or could have diabetes ??

The quote was indicated by WBCHCs as in the screening tool the questions guide them on the signs and symptoms that relate to Diabetic Mellitus which might assist them in thinking that the patient might have diabetic mellitus (see screening tools attached as supplementary files).

Discussions

Scott, Beckham, Gross, Pariyo, Rao, Cometto & Perry 2018, can be shortened to Scott et al 2018.

Citation shortened to Scott, Beckham, Gross et al 2018 as it was used for the first time.

Are the CHWs employed by the ward and supervised by a qualified nurse?

Ward based community health care workers are employed by the government under the department of Health and supervised by qualified professional nurses as they are attached to the primary health care facility. The name ward is political demarcation of villages that fall under an elected counsellor, Line 58.

Reviewer 2 Report

Comments and Suggestions for Authors

Introduction: The purpose of an introduction is to present the audience with general information and provide a reason as to why there is a need to conduct a study. To begin, the author failed to start the introduction with a powerful sentence about the study. Additionally, there are several inconclusive thoughts and vague language throughout. In line 16, it states that “non-probability purposive sampling method was used” but we are not given any context as to what this is. In order to strengthen this study, it would have been insightful to define the Ward Based Community Health Care Workers and how they will play a role in the study.

A major failure in this study is the presentation of noncommunicable disease screening tools. The author never defines what noncommunicable screening tools are being used which would provide more context to the study. Furthermore, since the author presents diabetes and hypertension as noncommunicable diseases being presented, it would have strengthened the manuscript to provide information about those disease states and the impact they have in South Africa. Lastly, the manuscript failed to present the literature gap of why there is a need to conduct this study in South Africa, in order for readers to be aware of the message the entire manuscript.

The manuscript has numerous grammatical errors present in the introduction. For example, in line 31 the word “organised” is misspelled as well as in line 34 “counselling” is misspelled.  

Methods: There are several flaws within the methodology. This section needs to be written better due to the lack of statistical analysis presented. Additionally, the author needs guidance on how to write the methods section. Furthermore, there is a lack of clarity due to not stating the results as a consequence of not providing the statistical analysis. 

Results: There are significant flaws in this section. For example, the authors did not state why a qualitative approach was selected and specifically interviews were used for the data collection. Since authors are confused between these two methods of data analysis content and thematic analysis and the significant flaws in these sections, this study cannot be accepted. The authors failed to use saturation process, which is a requirement for a qualitative study to be conducted. Furthermore, the lack of references is a significant concern for this manuscript.

The subthemes that were presented contained redundant material making it difficult to see other aspects of why these tools are essential. For example, in each subtheme, the author mentions the screening tools identified something, but it is never stated what tools led to those findings. Additionally, the quotations in the subthemes do not depict the message they are trying to convey.

Discussion: This section contains major concerns. A flaw of this study is that the objective states that screening households will decrease disease burden by quadruple diseases even though diabetes and hypertension were the only disease states mentioned in this manuscript. Additionally, this study claims to show how important using noncommunicable tools is in helping those in the community, but no one ever followed up to determine if disease burdens were reduced or if risk factors were reduced by using these noncommunicable tools to provide resources for patients.

Conclusion: Since the study does not adequately present the results, the conclusion cannot be fully supported.

Comments on the Quality of English Language

The manuscript has numerous grammatical errors present in the introduction. For example, in line 31 the word “organised” is misspelled as well as in line 34 “counselling” is misspelled.  

Author Response

Good day

Thank you for your time to review our paper. Kindly receive the report and the uploaded revised manuscripts.

AUTHOR’S CORRECTIONS TO REVIEWER 2 ‘S COMMENTS

Date: 31/01/2024

Title: Importance of utilizing noncommunicable diseases screening tools; Ward based community health care workers of South Africa Explains

Reviewers’ comments

Author’s corrections

Abstract

In line 16, it states that “non-probability purposive sampling method was used” but we are not given any context as to what this is.

Comment well received and corrected. In addition, full explanation was indicated in the sampling section of the same paper, line 125-131.

Introduction

-Failure to introduce the topic in general

-Defining non-communicable screening tool

-it would have strengthened the manuscript to provide information about those disease states and the impact they have in South Africa.

-Lastly, the manuscript failed to present the literature gap of why there isa need to conduct this study in South Africa,

-in line 31 the word “organised” is misspelled as well as in line 34 “counselling” is misspelled.

Suggestion welcomed, the introduction was restructured, and the manuscript was edited.

-Line 21-40 presented the general information related to the state of the problem or its impact.

-Line 83 had the role of screening tool to cover definition.

-The gap which prompted the researchers to conduct the study was indicated line 95-99

-Organised and counselling corrected to UK English though in south Africa, that’s how it is spelled.

Methods: There are several flaws within the methodology. This section needs to be written better due to the lack of statistical analysis presented.

More clarity on the methods was provided, however, the use of statistical analysis was not suitable for this paper as it is a qualitative study.

Results: There are significant flaws in this section. For example, the authors did not state why a qualitative approach was selected and specifically interviews were used for the data collection. Since authors are confused between these two methods of data analysis content and thematic analysis and the significant flaws in these sections, this study cannot be accepted. The authors failed to use saturation process, which is a requirement for a qualitative study to be conducted. Furthermore, the lack of references is a significant concern for this manuscript.

The subthemes that were presented contained redundant material making it difficult to see other aspects of why these tools are essential. For example, in each subtheme, the author mentions the screening tools identified something, but it is never stated what tools led to those findings. Additionally, the quotations in the subthemes do not depict the message they are trying to convey.

Thank you for your concern, however, this is purely a qualitative study where Focus Group Discussion interviews were conducted with participants.

The rationale of using qualitative approach was explained under study design line 105-107 and it cannot appear on the results as this section is meant for presentation of findings not methods.

-Content analysis was clarified line 148-157  

-Line 99-100 indicated the issue of data saturation and additional information about the process was added line 135-139.

-Additional references were added.

-Thank you for your concern however, redundant is a strong word to use, this study was conducted in 2022, however, the paper was subjected to editing and researchers relooked at the quotes to depict the message it was intended too.

Discussion: This section contains major concerns. A flaw of this study is that the objective states that screening households will decrease disease burden by quadruple diseases even though diabetes and hypertension were the only disease states mentioned in this manuscript. Additionally, this study claims to show how important using noncommunicable tools is in helping those in the community, but no one ever followed up to determine if disease burdens were reduced or if risk factors were reduced by using these noncommunicable tools to provide resources for patients.

-Researchers relooked at the discussion and Quadruple was removed.

-Thank you for this comment, however the study only focuses on the importance of utilizing the screening tools. Line 97-99 showed the number of patients seen daily and diagnosed monthly being referred by WBCHCWs.

Conclusion: Since the study does not adequately present the results, the conclusion cannot be fully supported.

This concern was addressed by relooking at the results section

Reviewer 3 Report

Comments and Suggestions for Authors

General: overall this is a well-written manuscript that clearly describes an interesting finding.  I have some specific comments, below.  In general, however I wonder if the topic of this research is suitable for the readership of IJERPH, as it does not address an environmental factor in any way.  Also as the aims and scope of the Journal state that all regional studies should be framed within a global context, I don’t see that the authors have addressed the global context of their research.  In fact on P 9 L 434, the authors say that the findings cannot be generalized to other settings as only three clinics were used.  I suggest that the authors revisit this point, and talk about what they have learned that could be generalized beyond the 3 clinics that were studied.

I leave final judgements on these points to the editors.

Specific:  P 2 L 63-64 states that Diabetic mellitus accounts for more than 40% of adults, with hypertension accounting for more than 50% of the condition globally, including South Africa (WHO 2017; NCD 2017). Please clarify, are the authors saying that the prevalence of diabetes mellitus is 40% among adults, and 50% are hypertensive?  These numbers seem high, or are these the rates for adults above a certain age or otherwise in a high risk category?  Please clarify.

P 2 L 68, is it possible to include a copy of the actual screening tools, maybe in an appendix or supplementary information to this manuscript?

P 2 L 84-85, please comment on how representative the population of these 3 clinics is on the larger population of the region, and Africa in general.

P 3 L 108, please describe the Colaizzi steps in data analysis and provide a reference.

P 4 L 150-151 states that “The use of screening tools guides the actions and workflow of WBCHCWs and improves the client’s compliance with the advice offered”.  I understand that the investigators can learn about the actions and workflow of the WBCHCWs from the interviews, but how did they gain any objective information about client compliance?  This needs to be explained and clarified. A similar comment on P 6 L 253 which states that “The current study found that family members change their lifestyle after being informed of the signs of chronic diseases, some seeking medical help.”  Is there any evidence that the investigators can cite to document this claim? 

 P 4 L 162 states that “…identify sick members of the unemployed…”, should this say “and the unemployed”?

P 6 L 278, indicates that some people who were screened were already prescribed some sort of medication, is there any data available about what portion of the people were prescribed such medication, but possibly not actually taking it?  And how the information provided through the screening can improve their compliance with the prescription?

P 9 L 417 the manuscript states that “According to the results of the current study, screening tools helped to provide patient education on how diabetic treatment ….” But the actual discussion that follows talks about the findings of McSharry et al, not the current investigation.

P 9 L 433, the authors describe this as an exploratory study, which seems to be accurate.  They should also talk more about other limitations, such as those I have raised in my specific comments.  Other limitations that should be mentioned include the fact that they don’t know what the rate of compliance was with the advice they provided through the screenings (that is what portion of people who received the advice actually followed up and sought health care, and of those who did obtain follow up, what portion were actually diagnosed with conformed diabetes or hypertension.  These limitations don’t render this study without value, but they should be mentioned as limitations. This would fit with the authors’ conclusion that the screenings can reduce the rate and burden caused by non- communicable diseases to society at large.

Author Response

Good day

Thank you for your time to review our paper. Kindly receive attached our corrections.

Regards

Round 2

Reviewer 2 Report

Comments and Suggestions for Authors

The methods section has major flaws. For example, the authors do not state why focus groups were selected as the methodology and the lack of references in this section make this section weak. 

It is recommended for the authors to review various studies on how to write qualitative results.

Rubin, H. J., & Rubin, I. S. (2011). Qualitative interviewing: The art of hearing data. sage.

Kitzinger, J. (1994). The methodology of focus groups: the importance of interaction between research participants. Sociology of health & illness16(1), 103-121.

Author Response

Good evening

Thank you for taking out your time to review our paper for the second time. All the comments were appreciated including the resources attached. Kindly receive our corrections.

Regards

Reviewer 3 Report

Comments and Suggestions for Authors

The authors have been responsive to earlier comments, recommend the revised manuscript be accepted and published. 

Author Response

Thank you for taking tome out to review our paper for the second time. We appreciated your efforts and for accepting that our paper be accepted for publication.

Regards